# Comparative Analysis of Egg Yolk Phospholipid Unsaturation and Its Impact on Neural Health in Alzheimer Disease Mice

**DOI:** 10.3390/foods14050792

**Published:** 2025-02-26

**Authors:** Yuhang Sun, Yao Wu, Bing Fang, Jingyu Li, Yue Liu, Haina Gao, Ming Zhang

**Affiliations:** 1School of Food and Health, Beijing Technology and Business University, Beijing 100048, China; 18339365328@163.com (Y.S.);; 2Key Laboratory of Precision Nutrition and Food Quality, Department of Nutrition and Health, China Agricultural University, Beijing 100083, China

**Keywords:** Alzheimer’s disease, egg yolk phospholipids, neuroprotective effects, lipidomics

## Abstract

The mechanism of egg yolk phosphatidylcholine (PC) in alleviating Alzheimer’s disease (AD) has not yet been clear. The fatty acid composition of PC, especially the ratio of polyunsaturated fatty acids (PUFA), may be a critical determinant of their structural and functional roles. This study aimed to conduct a comparative analysis of the unsaturation levels of egg yolk PC and their impact on neurological health in a murine model of AD. The results showed that oral administration of high and low unsaturation PC (HUP, LUP) enhanced learning and memory abilities in AD mice, with the HUP intervention demonstrating superior efficacy compared to the LUP. Follow-up biochemical analysis of the brain tissue also suggested that HUP intervention effectively mitigated oxidative-stress damage and inhibited tau hyperphosphorylation in AD mice. Meanwhile, lipidomic analyses of the mouse hippocampus revealed that HUP intervention substantially increased the levels of phospholipids, such as PEt (phosphatidylethanol) and BisMePA (bis(methylthio)phenylacetic acid), which are recognized as vital components of neuronal cell membranes. Furthermore, HUP intervention markedly elevated the levels of phospholipids incorporating PUFAs in the hippocampus. These results revealed a mitigating role for unsaturated egg yolk PC in AD prevention and offer new insights into AD prevention from a lipidomic perspective.

## 1. Introduction

Alzheimer’s disease (AD) is a complex neurodegenerative disorder with multiple pathological factors [1]. It is characterized by memory decline, cognitive impairment, and behavioral changes, which significantly affect daily life [2]. The prevention of AD through dietary patterns has garnered significant attention in recent years. The most studied dietary patterns in this context were the Mediterranean diet and the Mediterranean-DASH (dietary approaches to stop hypertension) diet, which has been associated with reduced oxidative stress and inflammation, both of which are implicated in the pathogenesis of AD [3,4,5].

Eggs are often found in many dietary patterns as an important food providing protein and lipid nutrients. Egg yolk phospholipids, natural bioactive substances with antioxidant and anti-inflammatory properties, hold the potential for the prevention and treatment of AD [6]. Chen et al. extracted and purified the main phospholipid component phosphatidylcholine (PC) from egg yolk and demonstrated its neuroprotective effect through a scopolamine-induced PC12 cell model. The results showed that pretreatment with PC from egg yolk could inhibit scopolamine-induced PC12 cell neurotoxicity and oxidative stress by downregulating acetylcholinesterase (AChE) activity, monoamine oxidase activity, and malondialdehyde (MDA) levels [7]. Similarly, Bao et al. found that egg yolk phospholipids exert neuroprotective effects by preventing hippocampal damage, regulating the activity of cholinergic biomarkers, maintaining normal acetylcholinesterase activity, and increasing the activity of scopolamine and oxidative stress-treated PC12 cells, thereby alleviating spatial memory deficits in model mice [8]. Egg yolk phospholipids play a key role in the development of the brain and neural networks by improving cognitive ability and participating in signal transduction [9]. The PC plays a crucial role in maintaining the stability and signal transduction of neuronal membranes, while phosphatidylethanolamine (PE) contributes to autophagy regulation and exhibits antioxidant properties [10]. However, the mechanism of action of egg phospholipids in alleviating AD is not yet clear, especially which of the specific types of phospholipids is the key component in ameliorating AD.

The fatty acid composition of phospholipids is a critical determinant of their structural and functional roles within biological systems. The fatty acid constituents of egg yolk phospholipids are diverse, comprising saturated, unsaturated, and polyunsaturated fatty acids (PUFAs) [11]. The phospholipids containing omega-3 and omega-6 PUFAs served as precursors for the production of eicosanoids and other lipid mediators that played significant roles in inflammation [12,13]. The remodeling of phospholipids, through processes such as the action of acyltransferases and phospholipase A2 enzymes, further influenced the availability of these fatty acids for conversion into bioactive compounds [14,15]. The dietary supplement with phospholipids rich in DHA (docosahexaenoic acid), a typical omega-6 PUFA, increased the number of dopaminergic neurons in MPTP-induced Parkinson’s model mice, thereby exerting neuroprotective effects [16]. Zhou et al. reported that phospholipids rich in omega-3 PUFA are beneficial for memory and cognition [17]. Nevertheless, saturated fatty acids may possess negative effects on cardiovascular and neurodegenerative diseases [18,19,20]. Consequently, egg yolk phospholipids enriched with unsaturated fatty acids were posited to possess enhanced neuroprotective properties. Consequently, determining which fatty acid composition of egg phospholipids is most effective in mitigating AD is of significant interest.

This study aimed to conduct a comparative analysis of the unsaturation levels of egg yolk PC and their impact on neurological health in a murine model of AD. Preventive intervention experiments were conducted using egg yolk PC with varying degrees of unsaturation. The study sought to identify the most effective fatty acid composition for alleviating AD by comparing behavioral assessments, the expression of key biomarkers in brain tissue, and lipidomic profiles in mice subjected to different PC interventions.

## 2. Materials and Methods

### 2.1. Extraction of PC

In this study, we collected 18 brands of eggs available in the Chinese market, all of which purportedly contain high levels of PUFA. Extraction and separation of PC was conducted as previously described in the literature [21]. In summary, five times the volume of 95% ethanol was added to the egg yolk, and the soluble fraction was collected and evaporated to dryness using a rotary evaporator to obtain crude phospholipid extracts. The crude phospholipids were then dissolved in trichloromethane to prepare a 1 mg/mL solution. Thin-layer chromatography was carried out using a solvent system comprising chloroform, methanol, and water in a ratio of 15:5:0.8 (*v*/*v*/*v*). Iodine vapor was employed to visualize the separated PC and PE. Subsequently, precooled ethyl acetate (1:5 *w*/*v*) at 4 °C was added to the crude extract to isolate the refined PC precipitate, which was then dissolved in 95% ethanol to prepare a 100 mg/mL solution for column purification. The refined PC samples were subsequently purified using chromatography on neutral aluminum oxide. The purity of the prepared egg yolk PC was assessed using UV spectrophotometry [22], revealing a purity level of 88.5%.

The saponification and methylation procedures were adapted from Hewavitharana et al. [23]. Lipid extract was refluxed with 8 mL of 2% NaOH in methanol at 80 °C, followed by 7 mL of 15% boron trifluoride methanol for 2 min. After cooling, 10–30 mL of n-heptane was added, shaken for 2 min, and saturated NaCl solution was used for phase separation. The upper n-heptane layer (5 mL) was transferred to a test tube with 3–5 g anhydrous Na_2_SO_4_, shaken, and settled for 5 min; the upper solution was collected in a sample vial for analysis.

### 2.2. Fatty Acid Analysis of PC

The analysis of fatty acids in PC was performed using gas chromatography, following the conditions specified by Yuan Nuo et al. [24]. The analysis was carried out on an SH-Rt-2560 capillary column (100 m × 0.25 mm, 0.20 μm) (RESTEK, Bellefonte, PA, USA) with helium as the carrier gas at a flow rate of 1.0 mL/min. The split ratio was set to 15:1, and the inlet temperature was maintained at 240 °C. The temperature program was as follows: initial column temperature of 130 °C for 5 min, then increased to 190 °C at 10 °C/min and held for 5 min, followed by a rise to 210 °C at 1 °C/min for 5 min, and finally to 240 °C at 4 °C/min for 5 min. Fatty acid quantification was performed using the external standard method, with the chromatographic peak areas used to calculate the fatty acid content in the samples [25].

### 2.3. Grouping and Administration of Animals

Five-week-old BALB/c mice, with an average weight of 20 ± 2 g, were procured from Beijing Huafukang Biotechnology Co., Ltd. (Beijing, China). The mice were housed in individual cages, with six animals per cage, under a controlled 12-h light/dark cycle and were provided with adequate feed and unrestricted access to water. Following a one-week acclimatization period, the mice were randomly divided into seven experimental groups: normal control (Control), Alzheimer’s disease model (Model), donepezil-positive control (Donepezil), low- and high-dose low-unsaturated egg yolk PC (LUP-L, LUP-H), and low- and high-dose high-unsaturated egg yolk PC (HUP-L, HUP-H). All groups, except the normal control group, received a daily oral gavage of 0.1 mL of aluminum chloride solution at a concentration of 4 mg/mL and a subcutaneous injection of 0.1 mL of D-galactose solution at 24 mg/mL into the neck region [26]. Concurrently, the donepezil-positive control group was administered a daily oral dose of 2.5 mg/kg of donepezil hydrochloride [27], whereas the egg yolk PC groups received oral doses of 1 g/kg/day (high dose) and 200 mg/kg/day (low dose) in 0.2 mL [8,28]. The normal and model groups received a daily oral gavage of 0.2 mL of physiological saline. This experimental protocol was approved by the Experimental Animal Committee of China Agricultural University (Approval No. AUJ21903202-5-1).

### 2.4. Behavioral Test—Morris Water Maze

The Morris water maze (MWM) test was conducted in accordance with the methodology outlined by Othman MZ et al. [29]. The circular MWM apparatus, measuring 120 cm in diameter and 40 cm in height, was segmented into four quadrants and filled with water maintained at 25 °C. An escape platform, 10 cm in diameter, was submerged 2 cm beneath the water surface. Following a four-day place navigation training, the frequency of platform crossings by the mice within a 60-s interval, along with the duration of stay and swimming distance in the third quadrant, were recorded.

### 2.5. Behavioral Test-Y-Maze New Arm

The experimental protocol adhered to the guidelines established by Kraeuter et al. [30]. An image acquisition system was employed to document the residence time and shuttle frequency of the mice in each quadrant over a five-minute testing period. This system also calculated and analyzed the frequency and duration of entries and stays in the novel quadrant to assess cognitive function.

### 2.6. Haematoxylin and Eosin (HE) Staining of Hippocampus

The experimental procedures followed Lin N et al.’s methodology with modifications [31]. After cardiac perfusion, mouse hippocampal tissues were fixed in 4% paraformaldehyde (pH 7.0), embedded in paraffin, and sectioned into 5-μm slices. The sections were stained with HE and examined under an optical microscope (CX40, Shunyu, Ningbo, China). Neuronal populations in the CA1 region were quantified using Image-Pro Plus 6.0 software (Media Cybernetics, Silver Spring, MD, USA).

### 2.7. Detection of Oxidative Stress Indicators in the Cerebral Cortex

The cerebral cortex was isolated from mouse brain tissue, and then lysate was added and sonicated for ten minutes in an ice water bath. Subsequently, the mixture was centrifuged at 10,000 rpm for 10 min, and the supernatant was collected and used with the reagent kit (Nanjing Jiancheng Bioengineering Institute, Nanjing, China) to detect oxidative stress indicators.

The detection method of MDA (malondialdehyde) (Nanjing Jiancheng Bioengineering Institute, Nanjing, China) is as follows: First, several centrifuge tubes were prepared with the following solutions: blank tube (OD_blank_): mix 0.1 mL anhydrous ethanol with working solution 1; standard tube (OD_st_): mix 10 nmol/L standard with working solution 1; test tube (OD_test_): mix 0.1 mL sample with working solution 1; control tube (OD_con_): mix 0.1 mL sample with working solution 2. All centrifuge tubes were mixed well and heated in a 95 °C water bath for 40 min. Afterward, the tubes were cooled under running water and then centrifuged at 4000 rpm for 10 min. Finally, the supernatant was collected and measured at a wavelength (Synergy H4, Biotek, Winooski, VT, USA) of 532 nm. The calculation formula is as follows:(1)$$\begin{array}{c} \mathrm{MDA}(\mathrm{nmol}/\mathrm{mgprot})\\ \;\;\;\;\;\;=\frac{{\mathrm{OD}}_{\mathrm{test}}-{\mathrm{OD}}_{\mathrm{con}}}{{\mathrm{OD}}_{\mathrm{st}}-{\mathrm{OD}}_{\mathrm{blank}}}\times \mathrm{Standards}\;(10\;\mathrm{nmol}/\mathrm{mL})\\ \;\;\;\;\;\;÷\mathrm{Tissue}\;\mathrm{protein}\;(\mathrm{mgprot}/\mathrm{mL}) \end{array}$$

The detection method of enzymatic activities of SOD (superoxide dismutase) is as follows: In a 96-well plate, add reagents as follows: Control well (OD_con_): Add 20 µL distilled water, 20 µL enzyme working solution, 200 µL substrate solution; Control blank (OD_con-blank_): Add 20 µL distilled water, 20 µL enzyme dilution, 200 µL substrate solution; Test well (OD_test_): Add 20 µL sample, 20 µL enzyme working solution, 200 µL substrate solution; Test blank (OD_test-blank_): Add 20 µL sample, 20 µL enzyme dilution, 200 µL substrate solution. Mix, incubate at 37 °C for 20 min, and measure absorbance at 450 nm. The calculation formula is as follows:(2)$$\mathrm{SOD}\left(\%\right)=\frac{{(\mathrm{OD}}_{\mathrm{con}}-{\mathrm{OD}}_{\mathrm{con}-\mathrm{blank}})-\left({\mathrm{OD}}_{\mathrm{test}}-{\mathrm{OD}}_{\mathrm{test}-\mathrm{blank}}\right)}{{(\mathrm{OD}}_{\mathrm{con}}-{\mathrm{OD}}_{\mathrm{con}-\mathrm{blank}})}\times 100\%$$(3)$$\mathrm{SOD}\left(U/mgprot\right)=\mathrm{SOD}\left(\%\right)÷50\%\times \frac{0.24\;mL}{0.02\;mL}÷\mathrm{Tissue}\;\mathrm{protein}\;(mgprot/mL)$$

The standard curve for SOD was prepared as follows: a 100 μg/mL standard stock solution was prepared using the SOD standard in a volumetric flask. The stock solution was then diluted with distilled water to obtain test solutions at concentrations of 50, 40, 25, 20, 10, 5, 4, and 2 U/mL.

The detection method of enzymatic activities of CAT (catalase) is as follows: Test tube: mix the sample with reagents 1 and 2, incubate at 37 °C for 1 min, then add reagents 3 and 4. Control tube: mix reagents 1 and 2, incubate at 37 °C for 1 min, then add the sample along with reagents 3 and 4. Mix thoroughly and measure absorbance at 405 nm using a microplate reader. The calculation formula is as follows:(4)$$\mathrm{CAT}\left(U/mgprot\right)=\mathrm{OD}\times 271÷0.1\;mL÷60÷\mathrm{Tissue}\;\mathrm{protein}\;(U/mgprot)$$

The detection method of enzymatic activities of GSH-PX (glutathione peroxidase) (Nanjing Jiancheng Bioengineering Institute, Nanjing, China) is as follows: Blank tube (OD_blank_): Add 1 mL standard solution and reagents 3 to 5. Standard tube (OD_st_): add 20 μmol/L GSH standard solution and reagents 3 to 5. Nonenzyme tube (OD_ne_): add the sample, 1 mL supernatant, and reagents 3 to 5. Enzyme tube (OD_e_): add 1 mL supernatant and reagents 3 to 5. The calculation formula is as follows:(5)$$\begin{array}{c} \mathrm{GSH}-\mathrm{PX}\left(U/mgprot\right)\\ \;\;\;\;\;\;=\frac{{\mathrm{OD}}_{\mathrm{ne}}-{\mathrm{OD}}_{e}}{{\mathrm{OD}}_{\mathrm{st}}-{\mathrm{OD}}_{\mathrm{blank}}}\times 20\;\mu mol/L\times 5÷5\;\;min\\ \;\;\;\;\;\;÷\left(0.2\;mL\right.\times \left.\mathrm{Tissue}\;\mathrm{protein}\;(mgprot/mL)\right) \end{array}$$

### 2.8. Western Blot (WB) Assay

Tissue homogenates of the hippocampus were subjected to Western Blot (WB) analysis to evaluate the expression levels of specific protein biomarkers. The samples were separated by SDS-PAGE electrophoresis and subsequently transferred onto a polyvinylidene fluoride (PVDF) membrane. Following blocking, the membrane was incubated overnight at 4 °C with the following antibodies: Phospho-tau (Ser404) specific antibody (Catalog No: 81383-1-RR, Proteintech) at a dilution of 1:10,000, tau polyclonal antibody (Catalog No: 10274-1-AP, Proteintech) at 1:10,000, phospho-tau (Ser202/Thr205) recombinant antibody (Catalog No: 82568-1-RR, Proteintech) at 1:10,000, GSK3β polyclonal antibody (Catalog No: 51065-1-AP, Proteintech) at 1:5000, and GAPDH polyclonal antibody (Catalog No: 10494-1-AP, Proteintech Group, Inc., Newark, DE, USA) at 1:5000. Subsequently, the membranes were incubated with horseradish peroxidase (HRP)-conjugated secondary antibodies at room temperature for 1 h. Following color development, the blots were scanned, and the gray values of the bands were quantified using ImageJ software (V1.8.0).

### 2.9. Lipidomics Analysis of Hippocampus

Six mice were randomly selected from each group, and 20 mg of hippocampal tissue was placed in a sterile centrifuge tube. Grinding beads and 400 μL of an extraction solution (methanol: water, 4:1, *v*/*v*) containing L-2-chlorophenylalanine were added. After centrifugation, the supernatant was collected and analyzed using Thermo Fisher’s UHPL-Q Exactive HF-X system (Thermo Fisher, Waltham, MA, USA) in positive and negative LC-MS modes. The MS and MS/MS data were processed with LipidSearch software (V5.1) to generate a data matrix containing retention time, peak area, mass-to-charge ratio, and metabolite identification. Spectral data were matched to a database for metabolite identification.

The data matrix was preprocessed by removing and imputing missing values, normalizing, and excluding variables with a relative standard deviation > 30% in quality control samples. Finally, the data matrix was standardized and the log transformed. Different lipid metabolites were identified using *t*-tests and OPLS-DA (*p* < 0.05, VIP-Oplsda > 1), followed by clustering and VIP binding analyses to explore their biological significance.

### 2.10. Data Processing

All the experimental data were expressed as mean ± SEM (Standard error of the mean). GraphPad Prism 9 software was used for statistical analysis. The comparison between two sets of data was conducted using a *t*-test, while the comparison between multiple groups was conducted using a one-way analysis of variance.

## 3. Results

### 3.1. Fatty Acid Composition of Egg Phospholipids

In this study, we collected 18 brands of eggs available in the Chinese market, all of which purportedly contain high levels of PUFA. Utilizing thin-layer chromatography, we determined that the crude extract of egg phospholipids was predominantly composed of PC, with minor quantities of PE and other phospholipids (Figure 1A). Subsequently, PC was isolated using alumina column chromatography, and its fatty acid composition was analyzed (Figure 1B). The analysis revealed that the saturated fatty acid content across the 18 PC samples varied from 46.7% to 60.1%, while the monounsaturated fatty acid (MUFA) content ranged from 11.6% to 28.8%, and the PUFA content ranged from 13.5% to 28.7%. Notably, PC derived from No. 4 exhibited the highest total unsaturated fatty acid content (MUFA + PUFA) at 53.3%, whereas PC derived from No. 17 had the lowest at 41.2%. A comparative analysis of these two samples indicated that the MUFA content in sample No. 4 was significantly higher than in sample No. 17. Particularly concerning docosahexaenoic acid (DHA), a crucial PUFA, sample No. 17 contained negligible amounts. For subsequent experiments, we selected PCs from samples No. 4 and No. 17 to represent categories of high and low unsaturation, respectively.

### 3.2. Impact of Unsaturation on PC to Improve Behavioral Test in AD Mice—Morris Water Maze

The Morris water maze test is a well-established neurobehavioral experimental technique employed to evaluate spatial learning and memory capabilities [29]. As illustrated in Figure 2A, during the initial two days of training, each group of mice required approximately 30 to 50 s to locate the platform. Beginning on the third day, mice in the AD group consistently located the platform in approximately 36 s, whereas the other groups exhibited continued improvement in their ability to locate the platform as training progressed. Figure 2B provides a quantitative analysis of the latency periods for each group of mice on the fifth day of training. Mice in the normal group demonstrated rapid platform location, with an average latency of 24.04 s. In contrast, the AD group exhibited a significantly longer average latency of 35.62 s (*p* < 0.05). Mice in the donepezil group showed a significantly reduced average latency of 24.82 as compared to the model group (*p* < 0.05). These findings confirmed the successful establishment of an AD model. Additionally, mice administered with LUP or HUP demonstrated significantly shorter average latencies than those in the AD group (*p* < 0.05), with mice in the high-unsaturation groups exhibiting superior performance.

Figure 2C illustrated representative swimming trajectories of mice from each experimental group on the sixth day of testing. Mice in the normal group, the positive drug group, and groups supplemented with LUP and HUP demonstrated efficient navigation to the target quadrant, successfully locating the original platform in the water maze. In contrast, the trajectories of mice in the AD group were erratic and predominantly along the periphery. As shown in Figure 2D,E, the model group exhibited significantly reduced target quadrant residence times and fewer platform crossings compared to the normal group (*p* < 0.05). Mice in the LUP and HUP groups, relative to the AD group, demonstrated increased target quadrant residence times and platform crossing frequencies. However, except for the HUP-H group, these differences were not statistically significant (*p* > 0.05). Furthermore, mice in the HUP group exhibited higher target quadrant residence times and platform crossing numbers compared to those in the LUP group. These findings suggested that characteristic egg yolk PC enhanced learning and memory abilities in mice, with the high-unsaturation PC demonstrating superior efficacy compared to the low-unsaturation PC.

### 3.3. Impact of Unsaturation on PC to Improve Behavioral Test in AD Mice—Y-Maze Test

The Y-maze test was conducted to assess the learning and memory capabilities of rodents following hippocampal injury, as determined by the exploration time and frequency in the new arm. As illustrated in Figure 3A, the exploration time of AD group mice in the new arm was significantly reduced compared to the normal group (*p* < 0.001), and the exploration frequency was also significantly decreased (*p* < 0.01). In contrast, intervention with donepezil in the positive control group significantly increased the exploration time of mice in the novel arm. Both HUP and LUP interventions also enhanced exploration time, with a more pronounced effect observed in the high unsaturated PC group (Figure 3A). Furthermore, HUP and LUP interventions significantly increased the exploration frequency (Figure 3B), with the most notable effect seen in the HUP-H group. These findings suggested that while both LUP and HUP significantly enhance learning and memory abilities in mice, HUP exhibits a more pronounced effect, highlighting the impact of the unsaturation level on the ameliorative effects of phosphatidylcholines on AD.

### 3.4. Impact of Unsaturation on PC to Alleviate Neuronal Damage in Mouse Hippocampus

The pathological morphology of the hippocampi of mice was evaluated through HE staining, as depicted in Figure 4. In the control group, neurons in the CA1 region of the hippocampus exhibited a tightly packed and orderly arrangement with distinct cellular outlines. Conversely, the AD group displayed a disorganized and loose neuronal arrangement, characterized by large intercellular gaps and signs of nuclear pyknosis and necrosis, leading to structural disruption and a decrease in cell count. Compared to the AD group, the hippocampal CA1 neurons in the donepezil and HUP groups closely resembled those in the normal group, maintaining normal cellular morphology (Figure 4A). The intercellular gaps were also reduced to some extent relative to the AD group. Furthermore, the number of neuronal cells was significantly diminished in the AD group, whereas interventions with donepezil, LUP, and HUP resulted in a notable increase in neuronal cell count, with the HUP group achieving effects comparable to the positive control drug, donepezil (Figure 4B). These findings suggested that highly unsaturated PC more effectively mitigated neuronal damage associated with AD.

### 3.5. Impact of PC Unsaturation on Oxidative Stress Indicators in the Cerebral Cortex of AD Mice

Oxidative products and antioxidant enzymes are frequently utilized as indicators to evaluate tissue peroxidation damage [32,33]. As illustrated in Figure 5A, MDA levels in the cerebral cortex of AD mice were significantly elevated compared to the control group (*p* < 0.05). Treatment with the donepezil resulted in a substantial reduction in cortical MDA levels. Interventions with HUP and LUP also reduced MDA levels, although only the HUP intervention group demonstrated a statistically significant decrease (*p* < 0.05). Regarding the three oxidoreductases SOD, CAT, and GSH-PX, significant reductions in CAT and GSH-PX levels were observed in the cerebral cortex of AD mice (*p* < 0.05), indicating increased oxidative damage. However, several intervention groups, including the donepezil group, did not exhibit significant improvements in the activity of these enzymes. These findings suggested that high unsaturation PC may mitigate oxidative stress damage in the AD mouse by decreasing MDA content; our results are consistent with those of previous studies [34,35].

### 3.6. Impact of PC Unsaturation on the Expression of GSK-3β and Tau Phosphorylation in the Hippocampus of AD Mice

Excessive phosphorylation of tau, which leads to the formation of neurofibrillary tangles, is recognized as a crucial pathological mechanism in AD [36]. GSK-3β is capable of phosphorylating tau at multiple residues, thereby contributing to tau hyperphosphorylation [37]. As illustrated in Figure 6A, B, the expression level of GSK-3β in the hippocampus of AD mice was significantly elevated compared to the control group (*p* < 0.05). However, treatment with the positive control drug, donepezil, markedly reduced GSK-3β levels in the hippocampus (*p* < 0.05). Among the intervention groups treated with HUP and LUP, only the high-dose HUP group demonstrated a significant downregulation of GSK-3β expression. Regarding tau protein expression, the findings indicated minimal variation across treatment groups; nonetheless, the high-dose HUP intervention group significantly decreased the expression of phosphorylated tau at serine 404 (*p*-tau S404) relative to the AD group (*p* < 0.05), achieving an effect comparable to that of donepezil. These results further implied that egg yolk PC, particularly with high unsaturated levels, may more effectively modulate the expression of key biomarkers GSK-3β and *p*-tau (S404) in the hippocampus, thereby mitigating the pathogenesis of AD.

### 3.7. Lipidomics Analysis of Mouse Hippocampal Tissue

The lipid composition of the hippocampus is integral to modulating cognitive functions and determining the degree of neural damage within the brain. Lipidomics, which involves a comprehensive analysis of lipid profiles, has emerged as a potent methodology for investigating these effects [38]. Figure 7A illustrates the various lipid types identified in the hippocampal tissues of all mice, along with the number of lipid molecules corresponding to each lipid type. Five principal lipid categories were identified: glycerophospholipids (GP), sphingolipids (SP), glycerides (GL), fatty acyls (FA), and sterols (ST). Notably, the GP molecules exhibited the greatest diversity and abundance. Within the specific subclasses, PC had the highest number of metabolites, followed by PE. 

Principal component analysis (PCA) was conducted to visually depict the overall distribution of samples and the variability both between and within the sample groups. As depicted in Figure 7B, the AD model group showed some overlap with the LUP group, whereas the HUP group demonstrated a clear distinction from the aforementioned groups. These findings suggested that the lipid composition of hippocampal tissues underwent more pronounced alterations following HUP intervention.

To find which lipid metabolite levels changed, we performed a volcano plot to visualize the differential lipid metabolites by considering both the VIP value (>1), *p*-value (<0.05) and fold change value (>1). As shown in Figure 7D, a total of 313 differential lipid metabolites were detected in the HUP group compared to the model group, with 205 upregulated and 108 downregulated. However, the LUP group only showed 92 differential lipid metabolites, with 46 upregulated and 46 downregulated (Figure 7C). The results presented above demonstrate that HUP treatment exerted a more pronounced impact on the lipid composition of hippocampal tissue compared to LUP treatment.

A hierarchical clustering analysis, combined with VIP analysis, was performed on the top 30 differential lipid metabolites. As illustrated in Figure 8A, within the top 30 metabolites exhibiting the highest VIP values in the LUP group relative to the AD group, LUP intervention resulted in an increased content of four unsaturated phospholipids, including PI (18:2/20:4) (phosphatidylinositol), PEt (16:1/18:1), PEt (18:0/22:4), and BisMePA (16:0/20:4). Conversely, HUP treatment led to an increase in the content of 17 unsaturated phospholipids, including 10 PEt (22:6/22:6, 20:1/18:1, 16:0/22:6, 18:0/18:2, 20:4/20:4, 18:1/22:5, 18:1/22:6, 16:1/18:1, 18:1/18:1, and 18:1/24:0), 5 BisMePA (18:1/22:6, 16:0/20:4, 16:0/22:6, 18:0/18:1, and 16:0/16:1), 1 PA (20:1/18:1) (phosphatidic acid), and 1 PMe (19:1/22:5) (phosphatidylmethanol), along with a significant increase in two triglycerides (TGs) (18:1/20:1/22:6 and 12:1/10:3/22:6). Notably, all 19 upregulated metabolites contained unsaturated double bonds, with 11 of these metabolites containing PUFAs (Figure 8B). These findings suggested that HUP intervention significantly enhanced the content of phospholipids, such as PEt and BisMePA, in hippocampal tissues and markedly increased the content of PUFAs in hippocampus lipids.

## 4. Discussion

Egg yolk PC holds promise for the prevention and treatment of AD. However, the precise mechanisms by which PC alleviates AD remain unclear. The fatty acid composition of PC, particularly the ratio of PUFA, may be a critical determinant of its structural and functional roles. This study aimed to conduct a comparative analysis of the unsaturation levels of egg yolk phospholipids and their impact on neurological health in a murine model of AD. The findings indicated that intervention with highly unsaturated PC was more effective in improving behavioral characteristics and mitigating neuronal damage in AD mice. This effect may be attributed to its enhanced capacity to ameliorate oxidative damage and reduce tau protein phosphorylation in brain tissues. Furthermore, lipidomic analysis of the mouse hippocampus revealed that the highly unsaturated PC intervention significantly increased the content of phospholipids, such as PEt and BisMePA, in hippocampal tissues and markedly elevated the levels of phospholipids incorporating PUFAs in hippocampus lipids.

Dietary intake of PC, a major phospholipid found in foods such as egg, soybean, and meat, has been shown to significantly influence human cognitive function. This effect is largely attributed to the fatty acid composition of PC, which may impact the cell membrane structure, neurotransmitter synthesis, and lipid metabolism. Phosphatidylcholine is a precursor to acetylcholine, a neurotransmitter crucial for memory and learning. The fatty acids attached to phosphatidylcholine molecules can vary, influencing their function and the overall impact on cognitive processes. For instance, DHA is often incorporated into PC and is known for its role in maintaining neuronal membrane fluidity and function. Studies have shown that higher levels of DHA in PC are associated with improved cognitive performance and a reduced risk of neurodegenerative diseases [39,40]. In addition to DHA, other PUFAs, like eicosapentaenoic acid (EPA) and arachidonic acid (AA), also play roles in cognitive health. EPA, another omega-3 fatty acid, has anti-inflammatory properties that may protect against cognitive decline, while AA, an omega-6 fatty acid, is involved in signaling pathways that can influence brain function [41,42]. In the current study, we selected two characteristic egg yolks from PC, representing the highest and lowest saturation levels, for subsequent experiments. Our findings corroborated previous conclusions, indicating that HUP (PUFA, 25.2%) significantly mitigated cognitive decline in mice, whereas LUP (PUFA, 13.5%) exhibited a comparatively lesser effect. Notably, there was minimal variation in MUFA content between LUP and HUP; however, the primary distinction arose in PUFA content, with LUP samples containing negligible DHA. The functional disparity between LUP and HUP appeared to be correlated with their respective DHA or PUFA content.

The lipid composition of the hippocampus plays a crucial role in influencing cognitive functions and the extent of nerve damage in the brain. Lipidomic analyses have revealed significant alterations in hippocampal lipid profiles in models of AD, suggesting that lipid dysregulation may contribute to the pathology of neurodegenerative diseases [38,43]. Studies have shown that diets rich in PUFAs can modulate hippocampal lipid composition, potentially offering protective effects against cognitive decline [44,45]. Lipidomic analyses of the mouse hippocampus revealed that HUP intervention substantially increased the levels of phospholipids, such as PEt and BisMePA, which are recognized as vital components of neuronal cell membranes. Furthermore, all 19 upregulated phospholipids contained unsaturated double bonds, with 11 phospholipids incorporating PUFAs. The presence of PUFAs in phospholipids is essential for maintaining membrane fluidity and facilitating the function of membrane proteins involved in signaling [46,47]. In contrast, the overall impact of the LUP intervention on the lipid composition in mice was relatively modest, with only two phospholipid fractions showing upregulation and no significant increase in unsaturation or polyunsaturated fatty acid (PUFA) levels. These findings suggested that the lipid composition of hippocampal tissues in AD mice experienced a more substantial positive effect following the intervention with highly unsaturated PC. This included an increase in the degree of unsaturation of phospholipids within hippocampal tissues, which, to some extent, influences membrane composition and affects cellular processes such as signal transduction and energy metabolism.

The observed increase in PUFA content in brain tissue may predict a reduction in oxidative stress, a critical factor in the onset and progression of AD [48]. Consequently, we measured the levels of MDA, SOD, CAT, and GSH-PX in the cerebral cortex, as these indicators are commonly employed to assess oxidative damage [35,49]. Qian L et al. demonstrated that phosphatidylcholine enriched with docosahexaenoic acid (DHA-PC) can mitigate liver oxidative stress, thereby offering protection against nonalcoholic fatty liver disease induced by a high-fat diet [34]. Aligned with previous research, our findings indicate that HUP intervention more effectively reduces MDA levels in the cerebral cortex and enhances the activities of SOD, CAT, and GSH, thereby exerting neuroprotective effects from an antioxidant perspective.

The excessive phosphorylation of tau, which leads to the formation of neurofibrillary tangles, is a key pathogenic mechanism in AD [36]. GSK-3β is known to phosphorylate tau at multiple residues, resulting in tau hyperphosphorylation [50]. Therapeutic strategies aimed at inhibiting GSK-3β may, therefore, be advantageous in the treatment of AD [51]. In this study, among the intervention groups treated with HUP and LUP, the high-dose HUP group exhibited a significant downregulation of GSK-3β expression. Furthermore, in terms of tau protein expression, the high-dose HUP intervention group significantly reduced the expression of phosphorylated tau at serine 404 (*p*-tau S404) compared to the AD group (*p* < 0.05). These findings suggested that high unsaturated PC may more effectively modulate the expression of key biomarkers GSK-3β and *p*-tau (S404) in the hippocampus, thereby mitigating the pathogenesis of AD.

## 5. Conclusions

In conclusion, oral administration of two different unsaturated levels of PC enhanced learning and memory abilities in AD mice, with the HUP intervention demonstrating superior efficacy compared to the LUP. Follow-up biochemical analysis of the brain tissue also suggested that HUP intervention effectively mitigated oxidative stress damage and inhibited tau hyperphosphorylation in AD mice. Meanwhile, lipidomic analyses of the mouse hippocampus revealed that HUP intervention substantially increased the levels of phospholipids, such as PEt and BisMePA, which are recognized as vital components of neuronal cell membranes. Furthermore, HUP intervention markedly elevated the levels of phospholipids incorporating PUFAs in hippocampus lipids. These results revealed a mitigating role for unsaturated level egg yolk PC in AD prevention and offer new insights into AD prevention from a lipidomic perspective.

## Figures and Tables

**Figure 1 foods-14-00792-f001:**
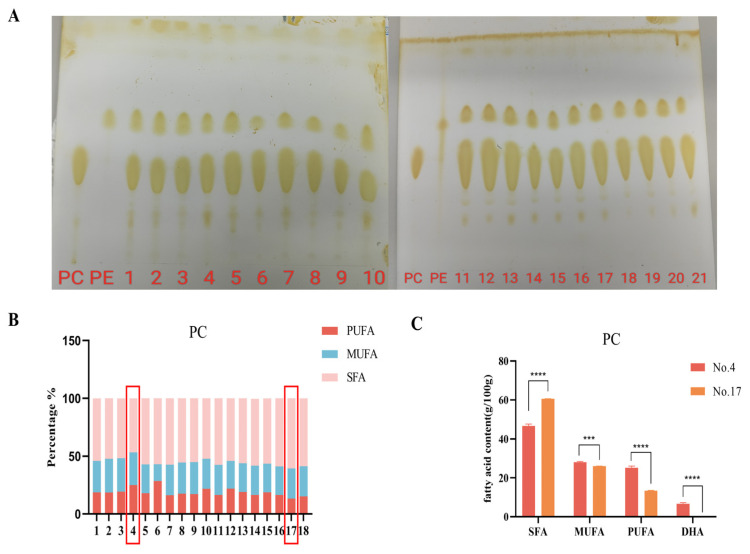
Fatty acid composition of egg phospholipids. (**A**) Thin layer chromatography results of egg yolk phospholipids from 18 brands of eggs; (**B**) Fatty acid composition of PC derived from 18 brands of eggs; (**C**) Comparison of highest and lowest unsaturated PCs (g/100 g). *** *p* < 0.001, **** *p* < 0.0001 represents a significant difference between the two groups. PC: phosphatidylcholine; PE: phosphatidylethanolamine; 1–21 represent the numbers of different egg brands respectively. The red boxes represent the two brands with the highest and lowest unsaturated fatty acids.

**Figure 2 foods-14-00792-f002:**
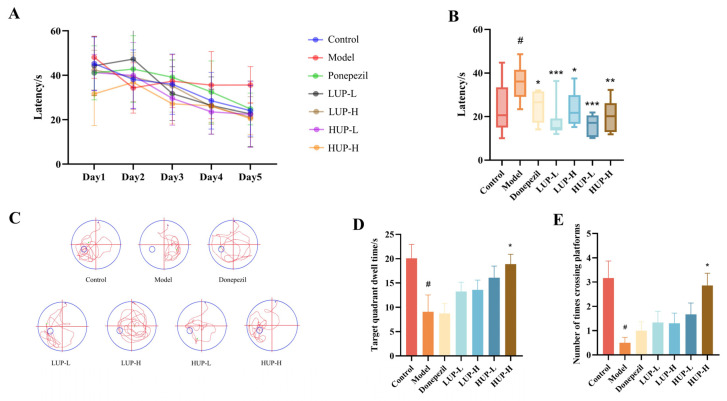
The effect of unsaturation on PC to improve behavioral test in AD mice—Morris water maze. (**A**) Changes in latency of mice in the first five days; (**B**) Quantitative latency plot of mice on the fifth day of testing; (**C**) Movement trajectory of mice on the sixth day of testing; (**D**) The residence time of mice within the target quadrant; (**E**) The number of times the mouse crossed the original platform area. # *p* < 0.05 represents significant difference versus control group; * *p* < 0.05, ** *p* < 0.01, *** *p* < 0.001 represent significant differences versus the model group.

**Figure 3 foods-14-00792-f003:**
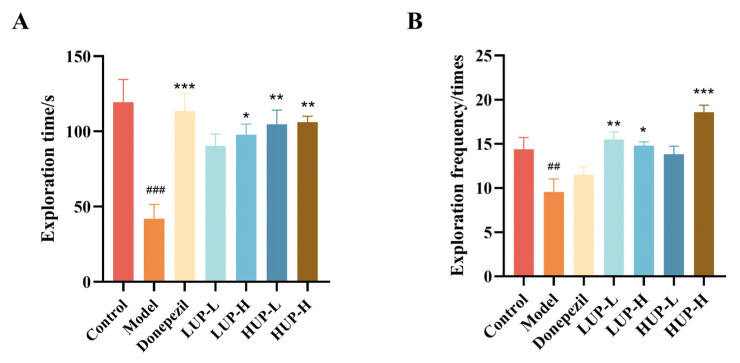
The effect of unsaturation on PC to improve behavioral test in AD mice—Y Maze. (**A**) Exploration time of each group of mice in the new arm; (**B**) The number of times each group of mice entered the new arm. ### *p* < 0.001 represents significant difference versus control group; ## *p* < 0.01 represents significant difference versus control group; * *p* < 0.05, ** *p* < 0.01, *** *p* < 0.001 represent significant differences versus the model group.

**Figure 4 foods-14-00792-f004:**
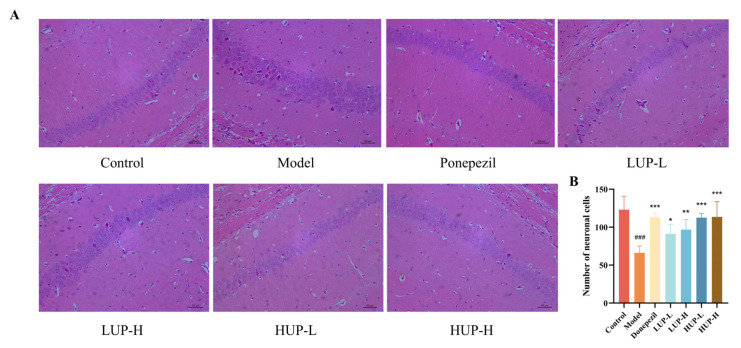
Impact of unsaturation on PC to alleviate neuronal damage in mice hippocampi. (**A**) HE stained sections of mice hippocampal neurons (magnification: 40 × 5; scale bar = 50 μm); (**B**) Mice hippocampal neuronal cell count. ### *p* < 0.001 represent significant difference versus control group; * *p* < 0.05, ** *p* < 0.01, *** *p* < 0.001 represent significant differences versus the model group.

**Figure 5 foods-14-00792-f005:**
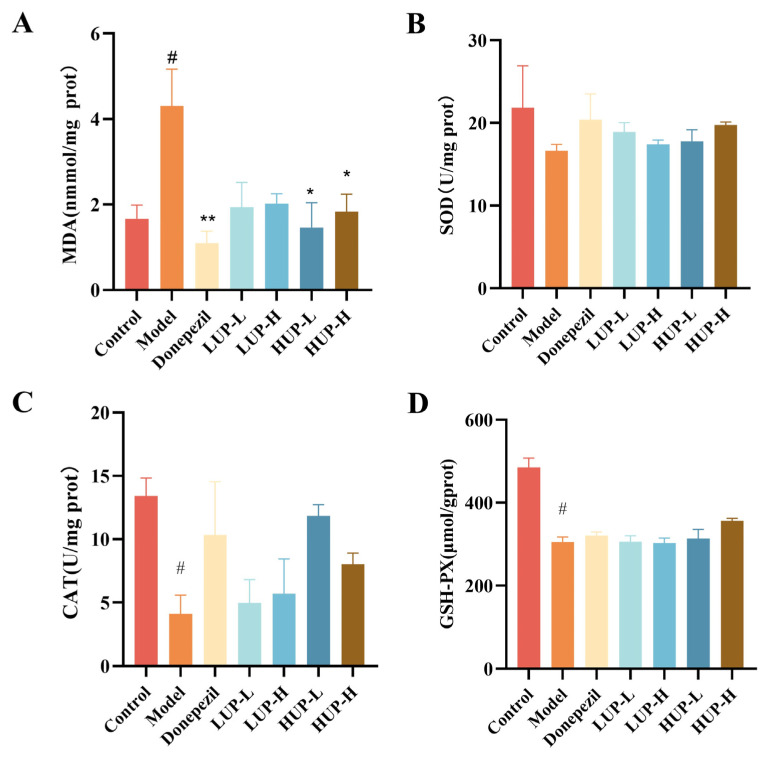
Impact of unsaturation on PC to alleviate oxidative stress in the cerebral cortex of AD mice. (**A**) MDA content; (**B**) SOD activity; (**C**) CAT activity; (**D**) GSH-PX activity. # *p* < 0.05 represents a significant difference versus the control group; * *p* < 0.05, ** *p* < 0.01 represent significant differences versus the model group.

**Figure 6 foods-14-00792-f006:**
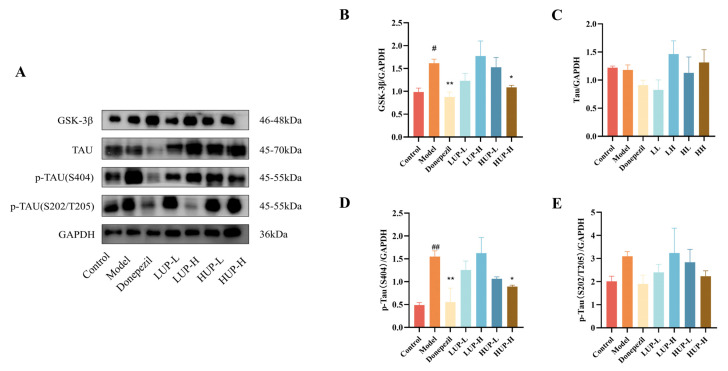
The expression of GSK-3β and tau phosphorylation in the hippocampus of AD mice. (**A**) Representative western blot; (**B**–**E**) The western blot assay of GSK-3β, tau protein, *p*-tau S404, and *p*-tau S202/T205. # *p* < 0.05, ## *p* < 0.01 represent significant differences versus the control group; * *p* < 0.05, ** *p* < 0.01 represent significant differences versus the model group.

**Figure 7 foods-14-00792-f007:**
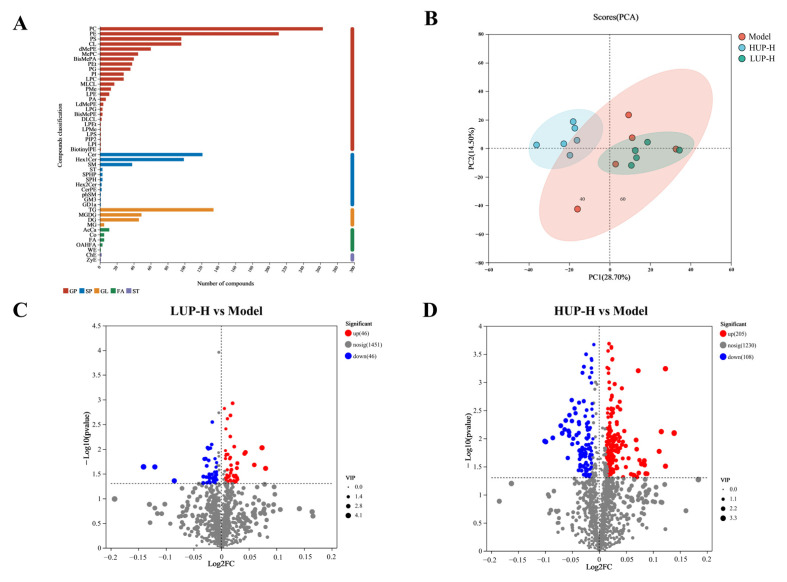
Lipidomic analysis of mouse hippocampal tissue. (**A**) Lipid profile of mouse hippocampal tissue.; (**B**) PCA analysis for HUP and LUP treatment groups; (**C**) Volcano plot of differential lipid metabolites between AD model and LUP group; (**D**) Volcano plot of differential lipid metabolites between AD model and HUP group.

**Figure 8 foods-14-00792-f008:**
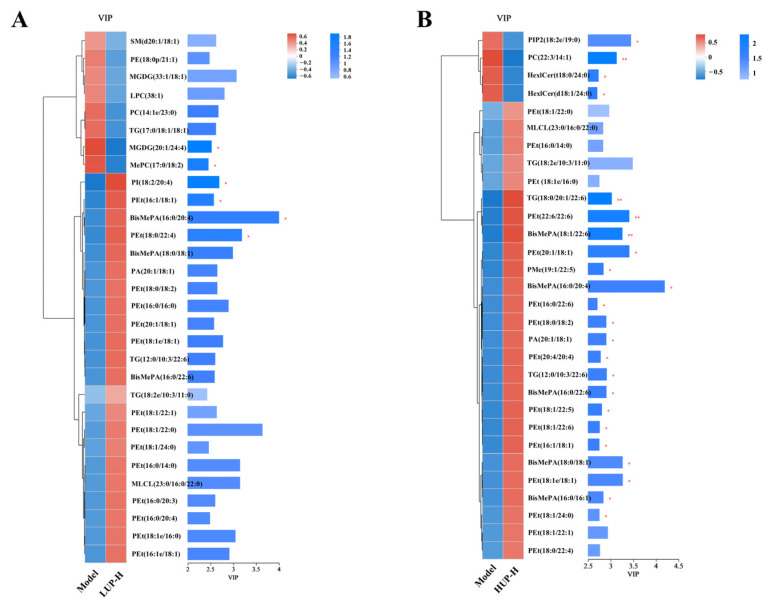
Hierarchical clustering combined with VIP analysis for representative differential metabolites. (**A**) Representative differential metabolites between LUP and AD model group; (**B**) Representative differential metabolites between HUP and AD model group, * *p* < 0.05, ** *p* < 0.01, represent significant differences versus the model group.

## Data Availability

The original contributions presented in the study are included in the article, further inquiries can be directed to the corresponding author.

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
