# Peer review of "Comparative Analysis of Egg Yolk Phospholipid Unsaturation and Its Impact on Neural Health in Alzheimer Disease Mice"

_foods, 2025, doi:10.3390/foods14050792_

Round 1

Reviewer 1 Report

Comments and Suggestions for Authors

Dear Authors,

The manuscript adheres to the Foods Instructions for Authors and provides a well-written introduction to the topic, along with a thorough discussion of well-planned experiments and strong conclusions. Overall, the manuscript is of high quality. However, the presentation of the results is somewhat limited by the low resolution and small size of the following figures: Fig. 2, Fig. 4B, Fig. 7, and Fig. 8. Please consider including higher-resolution images.

In addition, there are a few minor suggestions:

1.      It would be beneficial to separate the extraction process and the analysis of the PC.

2.      A more detailed explanation of the methods used, such as saponification and HE staining, would improve the reproducibility of the study.

3.      What was the unit of measurement for fatty acid content in Fig. 1?

4.      What was the magnification of the images in Fig. 4A?

5.       add a reference to support the following statement: “Aligned with previous research, our findings indicate that HUP intervention more effectively reduces MDA levels in the cerebral cortex and enhances the activities of SOD, CAT, and GSH, thereby exerting neuroprotective effects from an antioxidant perspective.”

Reviewer 2 Report

Comments and Suggestions for Authors

MANUSCRIPT 3478869_V1

TITLE: Comparative Analysis of Egg Yolk Phospholipid Unsaturation and Its Impact on Neural Health in Alzheimer's disease Mice

The manuscript 3478869_V1 “Comparative Analysis of Egg Yolk Phospholipid Unsaturation and Its Impact on Neural Health in Alzheimer's disease Mice”, presents an interesting study evaluating the impact on Alzheimer's disease in rats taking into account the difference in composition of unsaturated phospholipids in eggs.

This work is well structured, well planned and the research is competently carried out.

Methodology was adequate for the research but is not completely or does not present the methodology of some results presented.

Literature cited is adequate and most of the papers cited are from the last five years.

Statistical analysis was performed and is complete.

Results and discussion are properly discussed.

Conclusions are presented according to the results obtained.

Regarding the manuscript presented my comments are below:

1. Section 2. Materials and Methods. The manuscript presents results on page 8 regarding the determination of MDA, SOD, CAT and GSH-PX for which the determination methodology was not presented. Please, to allow the replication of the study, it is recommended to describe in detail all different methods for each parameter determined, including in all of them the complete description of the methodologies. The description must be complete indicating for example and it is necessary to describe the methodologies for determining the parameters separately and the amount of sample used in each parameter, the controls used, equations for the determination of parameters, calibration curves with standards used, conditions of operation with the instruments, among others.

2. Section 2. Materials and Methods – Please, in this section and all subsections consider adding the for each instrument used, model, producer and its location (Instrument Model, Producer, City, Abbr. State, Country). Proceed in the same way for all instruments used.

3. The manuscript contains many abbreviations and not all of them are written in full in advance. It is recommended that whenever you write an abbreviation for the first time, you either write out what it means in full or present a list of abbreviations in alphabetical order in your manuscript. Please review the entire manuscript as there are many abbreviations without their meaning being identified such as: DASH, SEM, DHA, SOD, CAT, GSH-PX, PI, PEt, BisMePA, PA and PMe.

4. Figures 7 and 8 - Please provide figures that can be read. It is recommended that the font size be increased and that the figures have a minimum resolution of 300DPI.

Round 2

Reviewer 1 Report

Comments and Suggestions for Authors

Dear Authors,

Thank you so much for incorporating my comments and suggestions, and likewise for the detailed explanations regarding the figures. The manuscript is improved, and I can truly see the thoughtful effort you've put into enhancing the clarity and quality of the work.

Author Response

Sincerely thank you for your very valuable advice.

Reviewer 2 Report

Comments and Suggestions for Authors

MANUSCRIPT 3478869_V2

TITLE: Comparative Analysis of Egg Yolk Phospholipid Unsaturation and Its Impact on Neural Health in Alzheimer's disease Mice

In the revised manuscript 3478869_V2 “Comparative Analysis of Egg Yolk Phospholipid Unsaturation and Its Impact on Neural Health in Alzheimer's disease Mice”, the authors present the new manuscript reformulated according to almost all the reviewers' recommendations.

Regarding the manuscript presented, I congratulate the authors for their effort in considering almost all reviewers' suggestions and for the valuable work presented. However, I have some recommendations that were not met and it is again recommended to authors to consider in the manuscript:

  1. Figures 7 and 8 – These figures are very difficult to read because the font size is very small. Please provide figures that can be read. It is recommended that the font size be increased in figures 7 and 8 and that the figures have a minimum resolution of 300DPI.
  2. Line 163 – Please indicate the chemical composition of working solution 1.
  3. Line 165 – Please indicate the chemical composition of the working solution 2.
  4. Lines 172 and 174 – please indicate what 20 μL enzyme working solution means, namely what the enzyme concentration of the enzyme working solution is.
  5. Lines 183 to 187 - Please indicate the chemical composition of reagents 1, 2, 3 and 4 and what volumes of sample and reagents are mixed.
  6. Lines 188 to 192 - Please indicate the chemical composition of reagents 3 and 5 and the standard solution and what quantities of sample and reagents 3 and 5 are mixed. Also indicate what the supernatant is and how the supernatant is obtained.

Author Response

Dear Editor and Reviewers,

We are grateful for your constructive comments and suggestions for our manuscript entitled " Comparative Analysis of Egg Yolk Phospholipid Unsaturation and Its Impact on Neural Health in Alzheimer's disease Mice " (Manuscript ID: foods-3478869). Your comments are very valuable and helpful for improving our paper, as well as the important guiding significance to our research. We have read the comments carefully and tried our best to make all the revisions clear. In the following, the responses to all the comments are provided one by one. We hope that the revised manuscript can satisfy the requirements for publication.

Responses to Reviewer#2

In the revised manuscript 3478869_V2 “Comparative Analysis of Egg Yolk Phospholipid Unsaturation and Its Impact on Neural Health in Alzheimer's disease Mice”, the authors present the new manuscript reformulated according to almost all the reviewers' recommendations.

Regarding the manuscript presented, I congratulate the authors for their effort in considering almost all reviewers' suggestions and for the valuable work presented. However, I have some recommendations that were not met and it is again recommended to authors to consider in the manuscript:

Comments 1:

Figures 7 and 8 – These figures are very difficult to read because the font size is very small. Please provide figures that can be read. It is recommended that the font size be increased in figures 7 and 8 and that the figures have a minimum resolution of 300DPI.

Response:Thank you for your careful review and suggestions. We have revised Figures 7 and 8 as follows: We have made font adjustments to the text in the image to ensure proper reading; And adjusted the resolution of the two images to above 300dpi.

Comments 2:

Line 163 – Please indicate the chemical composition of working solution 1.

Response:Thank you for your careful review of the Materials and Methods section. Regarding your inquiry about the chemical components of the reagents, we contacted the manufacturer (Nanjing Jiancheng Bioengineering Institute) and were informed that the specific composition is proprietary information protected by confidentiality agreements and thus cannot be disclosed. However, the manufacturer provided the complete kit instruction.

We have uploaded a detailed “Malondialdehyde (MDA) Assay Kit Instruction” in this response. Please check it out.

Comments 3:

Line 165 – Please indicate the chemical composition of the working solution 2.

Response:Thank you for your careful review of the Materials and Methods section. Regarding your inquiry about the chemical components of the reagents, we contacted the manufacturer (Nanjing Jiancheng Bioengineering Institute) and were informed that the specific composition is proprietary information protected by confidentiality agreements and thus cannot be disclosed. However, the manufacturer provided the complete kit instruction.

We have uploaded a detailed “Malondialdehyde (MDA) Assay Kit Instruction” in this response. Please check it out.

Comments 4.

Lines 172 and 174 – please indicate what 20 μL enzyme working solution means, namely what the enzyme concentration of the enzyme working solution is

Response:Thank you for your careful review of the Materials and Methods section. Regarding your inquiry about the chemical components of the reagents, we contacted the manufacturer (Nanjing Jiancheng Bioengineering Institute) and were informed that the specific composition is proprietary information protected by confidentiality agreements and thus cannot be disclosed. However, the manufacturer provided the complete kit instruction.

We have uploaded a detailed “Superoxide Dismutase WST-1 Assay Kit” in this response. Please check it out.

Comments 5.

Lines 183 to 187 - Please indicate the chemical composition of reagents 1, 2, 3 and 4 and what volumes of sample and reagents are mixed.

Response:Thank you for your careful review of the Materials and Methods section. Regarding your inquiry about the chemical components of the reagents, we contacted the manufacturer (Nanjing Jiancheng Bioengineering Institute) and were informed that the specific composition is proprietary information protected by confidentiality agreements and thus cannot be disclosed. However, the manufacturer provided the complete kit instruction.

We have uploaded a detailed “Catalase (CAT) Assay Kit” in this response. Please check it out.

Comments 6.

Lines 188 to 192 - Please indicate the chemical composition of reagents 3 and 5 and the standard solution and what quantities of sample and reagents 3 and 5 are mixed. Also indicate what the supernatant is and how the supernatant is obtained.

Response:Thank you for your careful review of the Materials and Methods section. Regarding your inquiry about the chemical components of the reagents, we contacted the manufacturer (Nanjing Jiancheng Bioengineering Institute) and were informed that the specific composition is proprietary information protected by confidentiality agreements and thus cannot be disclosed. However, the manufacturer provided the complete kit instruction.

We have uploaded a detailed “SOD (WST-1) Assay Kit Instruction” in this response. Please check it out.
